# EXPLAINING TEMPORAL GRAPH MODELS THROUGH AN EXPLORER-NAVIGATOR FRAMEWORK

**Wenwen Xia**[1,*]**, Mincai Lai**[2]**, Caihua Shan**[3,†]**, Yao Zhang**[4]**, Xinnan Dai**[2]**, Xiang Li**[5]**, Dongsheng Li**[3]
[1]Shanghai Jiao Tong University, [2]ShanghaiTech University, [3]Microsoft Research Asia
[4]Fudan University, [5]East China Normal University
`xiawenwen@sjtu.edu.cn`, {`laimc, daixn`}`@shanghaitech.edu.cn`
`yaozhang@fudan.edu.cn`, `xiangli@dase.ecnu.edu.cn`
{`caihuashan, dongsheng.li`}`@microsoft.com`

## ABSTRACT

While Graph Neural Network (GNN) explanation has recently received significant attention, existing works are generally designed for static graphs. Due to the prevalence of temporal graphs, many temporal graph models have been proposed, but explaining their predictions still remains to be explored. To bridge the gap, in this paper, we propose a Temporal GNN Explainer (T-GNNExplainer) method. Specifically, we regard a temporal graph as a sequence of temporal events between nodes. Given a temporal prediction of a model, our task is to find a subset of historical events that lead to the prediction. To handle this combinatorial optimization problem, T-GNNExplainer includes an explorer to find the event subsets with Monte Carlo Tree Search (MCTS), and a navigator that learns the correlations between events and helps reduce the search space. In particular, the navigator is trained in advance and then integrated with the explorer to speed up searching and achieve better results. To the best of our knowledge, T-GNNExplainer is the first explainer tailored for temporal graph models. We conduct extensive experiments to evaluate the performance of T-GNNExplainer. Experimental results demonstrate that T-GNNExplainer can achieve superior performance with up to ∼50% improvement in Area under Fidelity-Sparsity Curve.

## 1 INTRODUCTION

Temporal graphs are highly dynamic networks where new nodes and edges can appear at any time. The input is usually regarded as a sequence of events (node $i$, node $j$, timestamp $t$), which means there is an interaction (edge) between node $i$ and $j$ at timestamp $t$. It is ubiquitous in many real-world applications, such as friendship in social networks (Pereira et al., 2018; Barrat et al., 2021), and user-item interactions in e-commence (Li et al., 2021c). Many applicable temporal graph models (e.g., Jodie (Kumar et al., 2019), TGAT (Xu et al., 2020), TGN (Rossi et al., 2020)) are proposed considering both time dynamics and graph topology. Compared with static GNNs, temporal graph models learn the representation of each node as a function of time and then predict future evolutions, e.g., which interaction will occur and what time node attributes change.

Despite the success, all these models are black boxes and lack transparency. It is opaque how information aggregates and propagates over a graph and how a prediction is affected by historical events. Human-intelligent explanations are critical for understanding the rationale of predictions and providing insights into model characteristics. Explainers could increase the trust and reliability of temporal graph models when they are applied to high-stakes situations, like fraud detection in financial systems (Wang et al., 2021b) and disease progression prediction in healthcare (Li et al., 2021a). Besides, explainers also help check and mitigate the privacy, fairness and safety issues in real-world applications (Doshi-Velez & Kim, 2017).

While currently there are no methods for explaining temporal graph models, some recent explanation methods (e.g., GNNExplainer (Ying et al., 2019), PGExplainer (Luo et al., 2020) and Sub-graphX (Yuan et al., 2021)) for static GNNs are the most related. They identify the important nodes,

---

*This work was done during Wenwen Xia's internship at MSRA.
†Corresponding author.

edges and subgraphs for predictions by perturbing the input of GNN models. Obviously, these models cannot be used to explain a well-trained temporal graph model, as they cannot capture the temporal dependency mixed with the graph topology.

Here we propose T-GNNExplainer, an instance-level model-agnostic explainer for temporal graph models. For any prediction of a *target event*, we aim to find out important events from *candidate events*, which lead to the model's prediction of occurrence (or absence) of it. The candidate events are previously occurred events satisfying spatial and temporal conditions: they are in the $k$-hop neighborhood based on the message passing mechanism, and their timestamps should be close to that of the target event.

Specifically, T-GNNExplainer takes the advantages of search-based and learning-based GNN explainers together. Generally speaking, a learning-based explainer is inductive to all the target events, and explaining a target event is very quick once trained. A search-based explainer searches for the best result for each target event, which is more specific but time-consuming. While in this work, T-GNNExplainer is designed as a MCTS process with a learned navigator. We pretrain a navigator in advance to learn the inductive relationship between a target event and its candidate events. Then we utilize MCTS to explore the best combination of candidate events given any new target event. The navigator helps to bias the search process, significantly reducing the search time and improving the performance.

We evaluate T-GNNExplainer on both synthetic and real-world datasets for two typical temporal graph models (TGAT and TGN). On synthetic datasets, we simulate temporal events by the multivariate Hawkes process and pre-defined event relation rules. The highly accurate explanations demonstrate that T-GNNExplainer can find an exact influential event set. Since we do not know the ground truth for real-world datasets, the fidelity-sparsity curve is adopted to evaluate the superiority of T-GNNExplainer compared with baselines. We further provide a case study on synthetic datasets to illustrate the practical events found by T-GNNExplainer and navigation information.

## 2 RELATED WORK

### 2.1 TEMPORAL GRAPH AND TEMPORAL GRAPH MODELS

Graphs can be divided into four types by temporal granularity: static graph, graph with time-weighted edges, discrete-time dynamic graph (DTDG) and continuous-time dynamic graph (CTDG) (Kazemi et al., 2020). The typical graph neural networks (e.g., GCN (Kipf & Welling, 2017), GAT (Veličković et al., 2018), GIN (Xu et al., 2018)) can be used for the former two types to learn the static node embeddings. DTDGs are sequences of static graph snapshots taken at intervals in time. CTDGs are more general and are represented as a sequence of timestamped events, including edge/node addition, deletion, and feature transformations. In this work, we consider temporal graphs as CTDGs and take a sequence of timestamped events as model input since CTDGs are mainstream dynamic graphs with the finest time granularity.

Instead of static node embeddings, temporal graph models are required to learn dynamic node embeddings. DeepCoevelve (Dai et al., 2016) used RNNs to update node embeddings when some nodes are involved in new events. Jodie (Kumar et al., 2019) added the time projection module to make node embeddings evolve over time. However, they lack a GNN-like aggregation from node neighbors, which leads to the staleness problem (i.e., some node embeddings are out of date (Rossi et al., 2020; Kazemi et al., 2020)). Thus, CoPE (Zhang et al., 2021) and TGAT (Xu et al., 2020) are proposed to utilize the message passing mechanism to update node embeddings by its own events and its neighbors' events. It has been demonstrated to improve expressive power. TGN (Rossi et al., 2020) is an up-to-date framework and claims that most previous models are its specific cases. We choose the state-of-the-art TGAT and TGN as target models to be explained in the paper.

### 2.2 GRAPH EXPLAINERS

One popular way of explaining static graphs is to study the output variations of well-trained GNN models with respect to different input perturbations (Yuan et al., 2020b). Intuitively, the output changes vastly when critical nodes, edges, or subgraphs are perturbed. There are mainly two approaches assigning importance scores to graph entities by perturbations: learning-based and search-

based. Learning-based methods (Luo et al., 2020; Shan et al., 2021; Vu & Thai, 2020) leverage node representations generated by the trained GNN and adopt a neural network to learn crucial nodes/edges. They are trained with multiple instances, i.e., learning inductive explanation characteristics for multiple ones. Besides, search-based methods (Yuan et al. (2021); Wang et al. (2021a)) utilize heuristic search algorithms with a score function (e.g., defined by Shapley value or causality) to find an important input subset. Their inference time is longer because the search space of each instance is different, and they need to explore feasible solutions one by one. There are also some works possessing intrinsically interpretable architectures (Han et al. (2020); Li et al. (2021b); Xiao et al. (2022)) or generating model-level explanations (Yuan et al. (2020a); Shin et al. (2022)). Most of the self-interpretable models seek to a sparse subgraph during forward computation supported by the attention mechanism (Han et al. (2020)) or other internal scores (Li et al. (2021b); Cui et al. (2021)). There is a concurrent work that also explains temporal graph models He et al. (2022). However, they use discrete snapshots of a temporal graph while we focus on continuous event streams. In this work, we mainly focus on instance-level post-hoc explanation methods since most of the temporal graph models in the literature are not designed with specific consideration of explanations.

## 3 PRELIMINARY

### 3.1 TEMPORAL GRAPH MODEL

Assume that the input of temporal graph models is a sequence of events $\mathcal{S} = \{e_1, e_2, \cdots\}$. Each $e_i = \{n_{u_i}, n_{v_i}, t_i, \text{att}_i\}$ means that the node $n_{u_i}$ and $n_{v_i}$ have an interaction (edge) at timestamp $t_i$ with edge attribute $\text{att}_i$. The $\text{att}_i$ could be the interaction feature, or an indicator to represent $e_i$ is edge addition/deletion. Further, $e_i$ could involve only one node, $\{n_{u_i}, \text{null}, t_i, \text{att}_i\}$, to represent a node-wise event (node addition/deletion, or node attribute change). These events $\mathcal{S}$ constitute a temporal graph $\mathcal{G} = (\mathcal{N}, \mathcal{S})$ where $\mathcal{S}$ can be regarded as timestamped edges and $\mathcal{N}$ are nodes involved in $\mathcal{S}$. Since $\mathcal{G}$ and $\mathcal{S}$ are mutually defined, we regard $\mathcal{G}$ as both a temporal graph and a set of events in the following.

We utilize the setting defined in (Kazemi et al., 2020) to unify different temporal graph models as an encoder-decoder framework. The encoder is to learn the dynamic embedding for each node over time, and the decoder utilizes the node embeddings for downstream prediction tasks, such as future edge prediction. Specifically, let $\mathcal{G}^i$ denote the graph constructed just before the timestamp $t_i$, i.e., containing the events $\{e_1, \cdots, e_{i-1}\}$ but excluding $e_i$. The encoder takes $\mathcal{G}^i$ as the input and obtains $Z^i$ where $Z^i_{n_*}$ is the current embedding of the node $n_*$ at timestamp $t_i$. The decoder constructs the loss by predicting whether an interaction between a pair of nodes happen at this timestamp (the positive sample is $e_i = \{n_{u_i}, n_{v_i}\}$ while negative samples are the remaining pairs). Thus, the decoder uses $Z^i$ to predict the logit/probability of a new event between any pair of nodes, computes the loss to backpropagate the gradients, and updates the model parameters.

$$\textbf{Encoder}(\mathcal{G}^i) \rightarrow Z^i \qquad \textbf{Decoder}(Z^i) \rightarrow \text{Logit/Probablity of Events} \rightarrow \text{Loss}$$

Let $f(\cdot)$ denote a well-trained temporal graph model including the encoder and the decoder to simplify the notation. $f(\mathcal{G}^i)[e_j]$ means that we use the encoder to compute $Z^i$ by $\mathcal{G}^i$ at timestamp $t_i$ and computes the logit/probability of event $e_j$ by leveraging $Z^i_{n_{u_j}}$ and $Z^i_{n_{v_j}}$.

### 3.2 PROBLEM FORMULATION

Given the sequence of events and a well-trained temporal graph model $f(\cdot)$, the temporal explainer explains why the model predicts an event $e_k$ would occur or not. Specifically, we aim to find out a subset of events $\mathcal{R}^k$ from all the previous events $\mathcal{G}^k$ to maximize the mutual information $MI(Y_k, \mathcal{R}^k)$. $Y_k$ is the original prediction decided by $f(\mathcal{G}^k)[e_k]$, which is 0 or 1. $\mathcal{R}^k$ determines the distribution of the new prediction by $f(\mathcal{R}^k)[e_k]$. When they are strongly dependent, $\mathcal{R}^k$ is regarded as a good explanation for the prediction of occurrence/absence of event $e_k$.

According to (Ying et al., 2019), maximizing the mutual information $MI(Y_k, \mathcal{R}^k)$ is equivalent to minimizing conditional entropy $H(Y_k|\mathcal{R}^k)$ because $H(Y_k)$ is a constant. Then we can transform $H(Y_k|\mathcal{R}^k)$ into a cross entropy loss based on (Farnia & Tse, 2016):

$$\min_{\mathcal{R}^k} - \sum_{c=0,1} \mathbb{1}(Y_k = c) \log P(Y_{\text{new}} = c|\mathcal{R}^k) \tag{1}$$

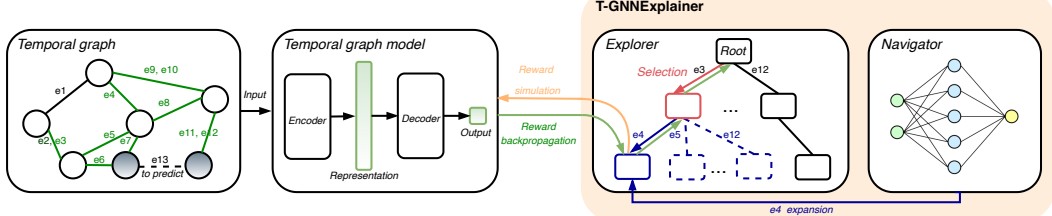

Figure 1: The framework of T-GNNExplainer. We first pre-train the navigator to learn the inductive relationship between events. Then we invoke the explorer to search out a specific combination of important events based on MCTS, including node selection, node expansion, reward simulation and backpropagation. When expanding a node, the navigator infers to decide which event is removed.

$P(Y_{\text{new}}|\mathcal{R}^k)$ is calculated by $f(\mathcal{R}^k)[e_k]$ and $c$ is the label indicating whether the event occurs or not.

**Objective.** Given a target event $e_k$ on which a prediction is made by $f(\cdot)$, T-GNNExplainer is a mapping $g$ to infer the important events $\mathcal{R}^k = g(e_k, \mathcal{G}^k, f(\cdot))$ by current temporal graph $\mathcal{G}^k$. T-GNNExplainer $g$ can also minimize the cross entropy loss for $K$ target events. Formally, the optimal explainer $g^*$ is defined as

$$g^* = \arg\min_{g} -\frac{1}{K} \sum_{k=1}^{K} \left[ \mathbb{1}(Y_k = 1) \log \sigma(f(\mathcal{R}^k)[e_k]) + \mathbb{1}(Y_k = 0) \log(1 - \sigma(f(\mathcal{R}^k)[e_k])) \right] \tag{2}$$

$$\text{subject to} \quad \mathcal{R}^k = g(e_k, \mathcal{G}^k, f(\cdot)) \text{ and } \mathcal{R}^k \subseteq \mathcal{G}^k \text{ and } |\mathcal{R}^k| \le N_r$$

Here we assume $f(\mathcal{R}^k)[e_k]$ is a single-dimensional logit value produced by the model $f(\cdot)$, and $\sigma(\cdot)$ indicates the sigmoid function. $\mathcal{R}^k$ should be concise so we use $N_r$ as a hyper-parameter to control the size of $\mathcal{R}^k$.

## 4 METHODOLOGY

### 4.1 OVERVIEW

Here we introduce how to identify an important subset of events $\mathcal{R}^k$ for the target event $e_k$. It is a combinatorial optimization problem where any subset of $\mathcal{G}^k$ whose size is equal to or smaller than $N_r$ could be the explanation. Besides, the search space of a temporal graph explainer is more significant than that of static graph explainers because of duplicate timestamped edges and node-wise events. Therefore, we design an explorer-navigator framework to effectively and efficiently obtain $\mathcal{R}^k$, shown in Fig. 1. The navigator is trained from multiple target events to capture the inductive correlation between events. The explorer is guided by the navigator and finds a more specific result based on Monte Carlo Tree Search. We will present them in the following.

### 4.2 NAVIGATOR

Inspired by previous parameterized explainers (Luo et al., 2020), we pretrain a navigator to provide a global understanding of the relationship among events. Concretely, the navigator is a feed-forward neural network $h_\theta(e_j, e_k)$, which infers the importance score of an event $e_j$ w.r.t a target event $e_k$. The scores will be leveraged in the node expansion of the explorer (Sec. 4.3) to facilitate the searching. The input features and the training/inference process are described as follows.

**Navigator features.** To capture the correlation between the target event $e_k = \{n_{u_k}, n_{v_k}, t_k, \text{att}_k\}$ and each candidate event $e_j = \{n_{u_j}, n_{v_j}, t_j, \text{att}_j\}$, we construct navigator input features as:

$$Z_{e_k, e_j} = [X_{n_{u_k}} || X_{n_{v_k}} || \text{Time}(t_k) || \text{att}_k || X_{n_{u_j}} || X_{n_{v_j}} || \text{Time}(t_j) || \text{att}_j]^T \tag{3}$$

$X$ represents the node feature matrix. $\text{Time}(\cdot)$ is a function converting a real-valued timestamp to a vector that could be learnable (Xu et al., 2020; Rossi et al., 2020) or not (Vaswani et al., 2017). In

this paper, we adopt the harmonic encoder (Xu et al., 2020) as the time function. We input all the candidate events w.r.t a target event as a batch into $h_\theta$.

**Training Process.** We use the same objective function described in Eq. 2. The output of $h_\theta$ is regarded as a soft-mask assigning weights to corresponding temporal edges of $\mathcal{G}^k$. If the model $f(\cdot)$ is differentiable w.r.t edge weights, we add the soft-mask as edge aggregation weights in the model, and then obtain the new prediction. Otherwise, the reparameterization trick is used in $f(\cdot)$. More details about reparameterization can be found in (Luo et al., 2020). Note that $h_\theta$ is trained inductively, i.e., events in the training set are different from those to be explained by T-GNNExplainer.

**Inference Process.** Provided with a candidate event $e_j$ and a target event $e_k$, we construct the input $Z_{e_k,e_j}$, put it into the trained navigator $h_\theta$ and infer the score, which will be utilized by the explorer.

### 4.3 EXPLORER

We adopt Monte Carlo Tree Search (MCTS) in the explorer. First of all, we initialize the root node as a set of candidate events. Then multiple rounds (a.k.a., rollouts) are conducted to expand nodes in the search tree, where each node represents a feasible subset of events in the search space. There are four aspects in each round: (1) select a path from the root to a leaf node; (2) expand new children by removing unimportant events according to the navigator in the path selecting procedure; (3) simulate reward of new nodes by temporal graph model; (4) backpropagate the leaf node's reward to update information in path nodes. At last, a node achieving the best reward and satisfying the sparsity threshold is our final explanation result. We present the pseudo-code in Appendix.

#### 4.3.1 INITIALIZATION

The root node includes all the candidate events, which are previously occurred events satisfying spatial and temporal conditions. Take the temporal graph in Fig. 1 as an example where we explain $e_{13}$. Assume that the encoder uses the 2-hop GNN-based aggregation, the set of seen events by encoder is $\{e_2, \cdots, e_{12}\}$. A temporal threshold is used to remove old events. If the threshold is set as 10, we preserve 10 recently occurred events. Finally, the root is initialized as $\{e_3, \cdots, e_{12}\}$.

#### 4.3.2 NODE SELECTION

We use $\mathcal{N}^i$ to represent a node in the search tree and use $e_j$ to indicate one action, i.e., discard the event $e_j$ from $\mathcal{N}^i$. We follow the UCT (Upper Confidence bound applied to Trees) formula proposed in (Kocsis & Szepesvári, 2006) to balance the exploitation and the exploration in the node selection. Assume we are on node $\mathcal{N}^i$, the action criteria is

$$e^* = \underset{e_j \in \mathcal{C}(\mathcal{N}^i)}{\arg\max} \left( \frac{c(\mathcal{N}^i, e_j)}{n(\mathcal{N}^i, e_j)} + \lambda \frac{\sqrt{\sum_{e_l \in \mathcal{C}(\mathcal{N}^i)} n(\mathcal{N}^i, e_l)}}{1 + n(\mathcal{N}^i, e_j)} \right) \quad (4)$$

$\mathcal{C}(\mathcal{N}^i)$ indicates the events already expanded in $\mathcal{N}^i$, $n(\mathcal{N}^i, e_j)$ is the count for selecting $e_j$ on node $\mathcal{N}^i$ in previous rollouts, and $c(\mathcal{N}^i, e_j)$ denotes the cumulative reward of selecting $e_j$ on node $\mathcal{N}^i$. The first component is exploitation: we select the node with a high average reward. The second component corresponds to exploration: we select the node with few simulations. We select and move to $\mathcal{N}^i$'s child node by removing the event $e^*$ from $\mathcal{N}^i$.

#### 4.3.3 NODE EXPANSION

The strategy of node expansion significantly influences the performance because it affects the search space and hence the best node's quality. Previous works expand all possible children for any selected node (Yuan et al., 2021). Instead, we only expand the best potential node to refine the search space. Assuming the selected node $\mathcal{N}^i$ is expandable (i.e., the number of children is less than the number of events contained in the node), the explorer invokes the navigator (Sec. 4.2) to obtain potential scores:

$$e^* = \underset{e_j \in \mathcal{N}^i / \mathcal{C}(\mathcal{N}^i)}{\arg\min} h_\theta(e_j, e_k) \quad (5)$$

$\mathcal{N}^i/\mathcal{C}(\mathcal{N}^i)$ means possible events which are not expanded in the previous rollouts. We remove the most unimportant event $e^*$ to expand a new node. Since the navigator is learned in advance and infers the score quickly, the additional cost is negligible. We could also expand the top-$k$ candidates or induce randomness to trade off the exploitation and exploration in the expansion step. For example, we select $e^*$ to expand with probability $1 - \epsilon$ and select a random unexplored $e_j$ with probability $\epsilon$.

The node selection and expansion are done alternatively. We start from the root node. We expand new child node(s) for the root according to Eq. 5. Then we choose the node with the highest value based on Eq. 4 from the root's original and new child nodes and move to it. Next we repeat the expansion and selection from the new node. The process ends when the current node is identified as a leaf node, e.g., the node has less than five events.

### 4.3.4 REWARD SIMULATION AND BACKPROPAGATION

The reward is simulated by the temporal graph model. In detail, we compute the reward of a leaf node $r(\mathcal{N}^{\text{leaf}})$ by computing the *negative* cross entropy loss [1] using Eq. 1, where $\mathcal{R}^k = \mathcal{N}^{\text{leaf}}$. In backpropagation, all the nodes $\mathcal{N}^i$ from root to the leaf will update $n(\cdot, \cdot)$ and $c(\cdot, \cdot)$ by adding the leaf node's reward and one respectively, i.e., $c(\mathcal{N}^i, e_l) = c(\mathcal{N}^i, e_l) + r(\mathcal{N}^{\text{leaf}})$, $n(\mathcal{N}^i, e_l) = n(\mathcal{N}^i, e_l) + 1$. $e_l$ is the action selected at $\mathcal{N}^i$.

## 5 EXPERIMENTS

In this section, we evaluate the performance of T-GNNExplainer with several baseline explainers. We first describe synthetic datasets, real-world datasets and target models in Sec. 5.1. Then we present the detailed experimental setup, including baselines and evaluation metrics in Sec. 5.2. Sec. 5.3 is a quantitative evaluation to demonstrate that T-GNNExplainer could surpass the baselines up to ∼50% improvement in the Area Under the Fidelity-Sparsity Curve (AUFSC). Furthermore, we investigate the navigator's effect in Sec. 5.4. Finally, a case study in Sec. 5.5 is constructed to show the explanations provided by T-GNNExplainer and navigation weights. The dataset statistics, target models' performance, and running time of all the methods are presented in Appendix. The code and datasets are attached in the supplementary.

### 5.1 DATASETS AND TARGET MODELS

**Real-world datasets**: We adopt two typical real-world temporal graphs: Wikipedia[2] and Reddit[3]. The Wikipedia dataset consists of ∼9300 active users and top edited pages and ∼160,000 temporal edges. A 172-dimensional user editing feature accompanies each temporal edge. The Reddit dataset is analogous to Wikipedia with ∼11,000 active users and subreddits and ∼700,000 temporal edges. The 172-dimensional temporal edge features come from user post contents.

**Synthetic datasets**: We utilize the Hawkes process (Hawkes, 1971) and tick library (Bacry et al., 2017) to generate synthetic datasets. In Fig. 2, we define four types of events ($E_1 \sim E_4$) in a graph. According to the multivariate Hawkes process, the intensity of an event is divided into two parts: endogenous and exogenous. For example, $E_0$ has endogenous intensity 0.5 to happen because of itself, and $E_3$ has exogenous intensity 2 influenced by the happening of $E_2$. Given a pre-defined event relation including endogenous/exogenous intensities, we adopt the tick library to simulate a sequence of events with timestamps. We generate two synthetic datasets with ∼ 10000 timestamps based on event relation v1 and v2 in Fig. 2. More details are described in Appendix.

**Target models**: We adopt two recent state-of-the-art temporal graph models TGAT (Xu et al., 2020) and TGN (Rossi et al., 2020). TGAT presents a temporal attention layer to aggregate a node's previous neighbours in chronological order and a time encoder to encode temporal information. TGN further proposes a memory store and update module to persist each node's temporal state. A point process model Transformer Hawkes Process (THP) is also compared in Sec. A.8 in Appendix. These models are trained in a self-supervised manner for real-world datasets, i.e., events seen on the graph are positive samples and randomly chosen unseen events are negative ones. For synthetic datasets,

---

[1]A larger reward indicates a better solution, hence negative CR is consistent with the objective in Eq. 2.

[2]http://snap.stanford.edu/jodie/wikipedia.csv

[3]http://snap.stanford.edu/jodie/reddit.csv

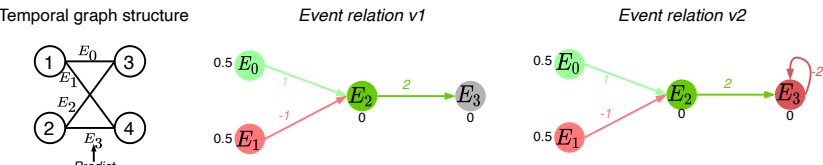

Figure 2: Synthetic temporal graph and pre-defined event relations. Values on nodes indicate endogenous intensities. Values on edges define exogenous intensities. Green edges are positive influences while red edges are negative influences. Grey edges reflect no influence.

happened $E_3$ timestamps are positive samples, and we uniformly sample random timestamps as negative samples.

## 5.2 BASELINES AND SETUPS

**Baselines**: We compare the performance of T-GNNExplainer with several baseline methods. (1) We implement PGExplainer (PG) in (Luo et al., 2020) and adapt it for the temporal graph scenario. The adapted PG computes a weight to each event instead of each edge. The input information for event $e_j$ is the same as the $Z_{e_i,e_j}$ in Eq. 3. We add the output score of PG to the attention weights in the target model for all layers and use the same training objectives as T-GNNExplainer. (2) For the attention-based explainer (ATTN), we extract the attention weights in TGAT/TGN and average the values over all layers. The averaged weights are regarded as importance scores. (3) Besides, we implement a straightforward explainer by perturbing one candidate event (PBONE), i.e., we compute the importance of each event $e_j \in \mathcal{G}^k$ by feeding $\mathcal{G}^k/\{e_j\}$ into the target model. Moreover, we also compare with the self-interpretable THP-based THPExplainer in Sec. A.8 in Appendix.

**Evaluation metrics**: We adopt the fidelity $Fid(f(\mathcal{G}^k)[e_k], f(\mathcal{R}^k)[e_k])$ and the sparsity $Sp(\mathcal{R}^k, \mathcal{G}^k)$ to evaluate the performance. Instead of using the difference between the original and new prediction probabilities to define the fidelity in the previous work (Yuan et al., 2020b), we use the difference between logits because logits could exhibit explainers' performance more clearly. The fidelity is defined as $Fid(f(\mathcal{G}^k)[e_k], f(\mathcal{R}^k)[e_k]) = \mathbb{1}(Y_k = 1)(f(\mathcal{R}^k)[e_k] - f(\mathcal{G}^k)[e_k]) + \mathbb{1}(Y_k = 0)(f(\mathcal{G}^k)[e_k] - f(\mathcal{R}^k)[e_k])$. Besides, sparsity is defined as $Sp = |\mathcal{R}^k|/|\mathcal{G}^k|$. The higher fidelity and higher sparsity mean a better result. We draw the fidelity-sparsity curve and compute area under the curve **AUFSC** to evaluate the performance. A larger AUFSC indicates a better performance. Note that AUFSC may be negative because fidelity could be negative.

Furthermore, we also use the metric **Best Fid**, indicating the best fidelity ever found by the explainer without sparsity limitations. For T-GNNExplainer, we traverse all tree nodes to find a node with the best fidelity. For baseline explainers, we rank all the candidate events in ascending order by their importance scores produced by the explainer and successively preserve top events to find the subset with the best fidelity.

**Experimental setup**: We use a two-layer MLP with 128 hidden units to instantiate the navigator $h_\theta$. We set the exploration parameter $\lambda$ to 5 and the rollout number to 500 in the explorer. Following the same setting in TGAT and TGN, we adopt a two-layer attention architecture and harmonic encoding for timestamps. We train both TGAT and TGN with a 70%, 15%, and 15% splitting scheme of datasets based on timestamps. For all methods, we limit the number of candidate events to 25 and randomly sample 500 events in the test dataset as target events for the explanation. We use a machine with an RTX 2080 GPU and a 48-core Intel(R) Xeon(R) CPU@2.2GHz. More hyper-parameters are listed in Appendix.

## 5.3 PERFORMANCE COMPARISON WITH BASELINES

In this section, we report the quantitative results in Table 1 and Table 2 for real-world and synthetic datasets respectively[4]. We find that T-GNNExplainer outperforms baseline explainers significantly and consistently for two metrics on all the datasets. On the real-world datasets, the gains of AUFSC (Best Fid) are up to 53%(26%), 86%(45%), 134%(47%), and 74%(50%) w.r.t to the leading baseline

---

[4]Best results are in bold and the second best are underlined.

Table 1: Best fidelity (↑) and AUFSC (↑) achieved by each explainer on real-world datasets.

| | Wikipedia | | | | Reddit | | | |
|---|---|---|---|---|---|---|---|---|
| | TGAT | | TGN | | TGAT | | TGN | |
| | Best Fid | AUFSC | Best Fid | AUFSC | Best Fid | AUFSC | Best Fid | AUFSC |
| ATTN | 0.891 | 0.564 | 0.479 | 0.073 | 0.658 | -0.654 | 0.575 | 0.289 |
| PBONE | 0.027 | -2.227 | 0.296 | -0.601 | 0.167 | -2.492 | 0.340 | -0.256 |
| PG | 1.354 | 0.692 | 0.464 | -0.231 | 0.804 | -0.369 | 0.679 | 0.020 |
| T-GNNExplainer | **1.836** | **1.477** | **0.866** | **0.590** | **1.518** | **1.076** | **1.362** | **1.113** |

Table 2: Best fidelity (↑) and AUFSC (↑) achieved by each explainer on synthetic datasets.

| | Synthetic v1 | | | | Synthetic v2 | | | |
|---|---|---|---|---|---|---|---|---|
| | TGAT | | TGN | | TGAT | | TGN | |
| | Best Fid | AUFSC | Best Fid | AUFSC | Best Fid | AUFSC | Best Fid | AUFSC |
| ATTN | 0.555 | 0.390 | 2.178 | 1.624 | 0.605 | 0.291 | 0.988 | -0.634 |
| PBONE | 0.044 | -2.882 | 0.000 | -3.311 | 0.096 | -4.771 | 0.320 | -5.413 |
| PG | 0.476 | -0.081 | 2.006 | 0.626 | 1.329 | -0.926 | 1.012 | -1.338 |
| T-GNNExplainer | **0.780** | **0.666** | **2.708** | **2.281** | **1.630** | **1.331** | **4.356** | **3.224** |

in four scenarios. ATTN and PG obtain comparable performance while PBONE performs the worst. The results of synthetic datasets are analogous to those of real-world datasets.

Besides, we illustrate the fidelity-sparsity curve on the real-world datasets intuitively, shown in Fig. 3. T-GNNExplainer achieves the highest final fidelity than other baselines and is also the highest one under a given sparsity threshold. Moreover, with a relatively small sparsity threshold, e.g., 0.2, T-GNNExplainer can already find a solution with a high fidelity compared to its final best value, which indicates that T-GNNExplainer explores the low-sparsity event subsets efficiently. Without the searching procedure, the fidelity of PG and ATTN increase slowly. PBONE performs the worst in all scenarios because it treats each event independently. More performance investigations of the navigator are illustrated in Sec. A.5 and Sec. A.6 in Appendix.

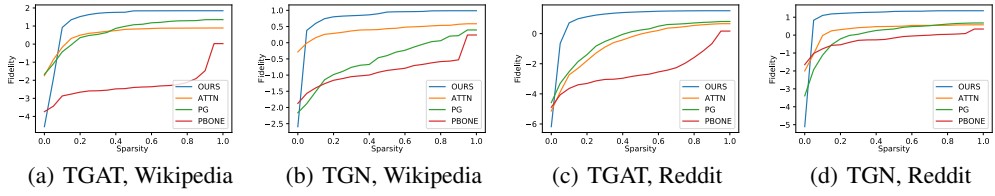

| (a) TGAT, Wikipedia | (b) TGN, Wikipedia | (c) TGAT, Reddit | (d) TGN, Reddit |

Figure 3: Fidelity-sparsity comparison of explainers on real-world datasets with target models.

## 5.4 EFFICIENCY INFLUENCE OF THE NAVIGATOR

In this section, we investigate the efficiency enhancement of the navigator. We set the rollout number to 500 for *with navigator* and *without navigator*. Here, we compare the time exhausted to achieve a fidelity threshold. Let the *best fidelity* denote the larger final fidelity between *with navigator* and *without navigator*. We set the fidelity threshold to $0.8 \times best\ fidelity$ to compare the time exhausted.

Results for all the datasets and both models are shown in Fig. 4. The results are averaged over all target events. We can find that the running time of *with navigator* is always less than that of *without navigator* under all settings. The efficiency improvement of *with navigator* is about 70.1%, 60.1%, 48.2%, and 28.1%. On the synthetic datasets, we can observe similar results as those on real-world datasets. The speedup of the navigator is about 83.85%, 96.43%, 78.48%, and 43.66% respectively under four synthetic settings. Overall, the navigator effectively speeds up the searching procedure of the explorer to achieve reasonable solutions. More runtime comparisons with baselines and complexity analysis of T-GNNExplainer are illustrated in Sec. A.4 in Appendix.

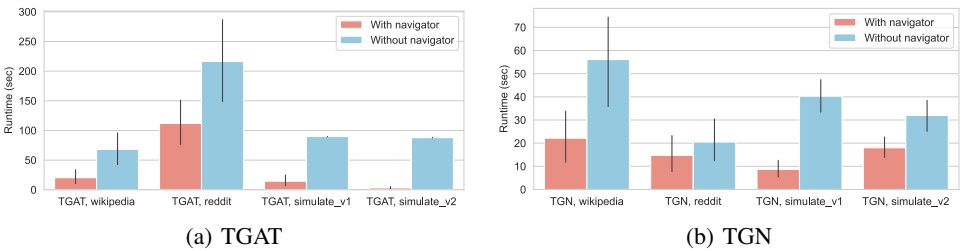

| (a) TGAT | (b) TGN |

Figure 4: Efficiency of the navigator on all the datasets with target models.

## 5.5 CASE STUDY

In this section, we visualize explanations for target events with TGAT on the synthetic dataset. Specifically, we show the scores given by T-GNNExplainer, the navigator, and the target models' intrinsic attention module. The values are normalized in [0, 1]. Since T-GNNExplainer returns a subset, we assign 1 to the selected events and 0 to others. The bar colours are consistent with type colours in Fig. 2. The light green and green colours represent positive events to trigger the event $E_3$, while the red colour indicate negative events to inhibit the event $E_3$. The grey are irrelevant events. Because we explain the happened target event $E_3$, a good explainer should assign relatively high scores to green bars and low scores to gray and red events.

Fig. 5 and Fig. 6 present the cases on synthetic v1 and v2. ATTN finds a dense event subset including irrelevant or negative events (Fig. 5(c)), or overlooks some previous positive events (Fig. 6(c)). The navigator obtains a better result than ATTN. It assigns high scores to green bars and almost eliminates the gray and red ones. Assisted by the navigator, T-GNNExplainer utilizes the explorer to further filter the events and make sure the final result is concise. Overall, our final event set is sparse while it remains the most important events leading to the occurrence of the target event $E_3$.

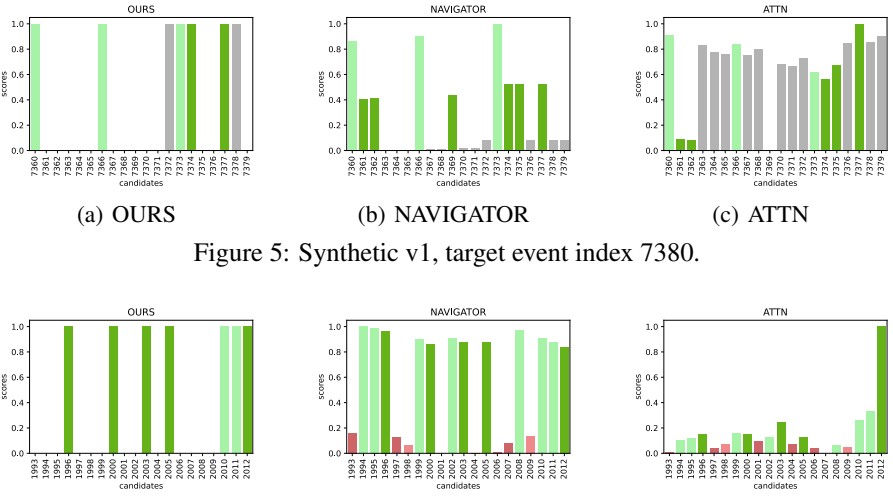

| (a) OURS | (b) NAVIGATOR | (c) ATTN |

Figure 5: Synthetic v1, target event index 7380.

| (a) OURS | (b) NAVIGATOR | (c) ATTN |

Figure 6: Synthetic v2, target event index 2013.

## 6 CONCLUSION

In this paper, we propose the T-GNNExplainer for temporal graph model explanation. We design a novel explorer-navigator framework to search for explanations effectively and efficiently. Experimental results illustrate the superiority of T-GNNExplainer on both synthetic and real-world datasets with two typical temporal graph models. Since instance-level explanations can be local and sensitive to the input, exploring model-level explanations for temporal graph models could be future work. Besides, how to generate simulated datasets to mimic real-world dynamic graphs more accurately also deserves further investigation.

## ACKNOWLEDGEMENTS

We thank Prof. Shenghong Li and our anonymous reviewers for delightful discussions and for providing feedback on our manuscript. This work is funded by the National Nature Science Foundation of China under Grant 61971283, Shanghai Municipal Science and Technology Major Project under Grant 2021SHZDZX0102, and Shanghai Pujiang Talent Program No. 21PJ1402900.

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
