# OpenReview forum: "Explaining Temporal Graph Models through an Explorer-Navigator Framework"
_ICLR.cc/2023/Conference — ICLR 2023 poster_

### Official Review · Reviewer_zSpG · 2022-10-23

**Confidence:** 3
**Correctness:** 3
**Technical Novelty And Significance:** 3
**Empirical Novelty And Significance:** 3
**Recommendation:** 6

**Clarity, Quality, Novelty And Reproducibility:**

Most of all, the proposed model is clear to understand the structure. The model seems to be quitely novel since it takes the two advantages of search- and learning-based GNN explainers, and is applied to under-explored area, temporal graphs. The experimental results with other baselines show that it can achieve superior performance than others, so it would be significant.

**Strength And Weaknesses:**

Strength
- The proposed model is well designed. The authors take the advantages of search- and learning-based GNN explainer together. More specifically, the authors apply search-based explainer to explorer for finding the best results and use MCTS to reduce search time. Moreover, they train the navigator to learn the inductive relationships.

- It can be a significant contribution since GNN explaining model for temporal graphs is under-explored.

- Overall, the paper seems to be well-organized. In my view, the authors carefully composed the paper to understand what GNN explainer is, what the proposed model is, and so on.

Weaknesses
- Several notations are confusing to me. For example, the authors reuse the notation G^i in section 4.3.2, with (minorly) different meaning. But it can be confusing for readers to follow the paper. The reviewer expects that the authors use clear notations to be understood more easily.

- Literature survey would not be enough. The authors claim that it is the first explainer for temporal graph models. But Z. Han et al., "Explainable subgraph reasoning for forecasting on temporal knowledge graphs" at ICLR 2021 tried to construct an explaining model for temporal knowledge graphs. In my view, although it is applied to temporal knowledge graphs, it needs to be mentioned and discussed in the related work section.

- I think that several experiments are more needed. As far as I can tell, many works for GNN explainers (e.g., D. Luo et al., "Parametrized explainer for graph neural network" at NeurIPS 2020) illustrated the performances, and so that there can be several illustration rather than figure 5 and 6 in this paper.

- Performance comparison between the proposed model with navigator and without navigator seems to be needed. The authors show the efficiency of the navigator with figure 4. But there is no performance (Fidelity, sparsity) comparison between them. If there is the performance comparison, the necessity of navigator can be more convinced.



**Summary Of The Paper:**

The paper "Explaining temporal graph models through an explorer-navigator framework" proposes a graph neural network (GNN) explainer model for temporal graphs. More specifically, the proposed model has mainly two components which are explorer and navigator. The explorer is to find the event subsets for explanation, and the navigator is to reduce the search space. The authors conduct various experiments to evaluate the performance and show that the proposed model can achieve superior performance than others.

**Summary Of The Review:**

In general, the paper seems to be well-organized and the proposed model can be significant. The idea of taking two advantages of search- and learning-based GNN explainer would be novel. But as aforementioned, 1) several notations need to be revised to be clearly understood, 2) enough literature survey is conducted and 3) More experiments and some illustrations seems to be needed for better version of paper.

---

> ### Author Response · Authors · 2022-11-18
> **Response to Reviewer 1, Q4**
>
> > Q4: Performance comparison between the proposed model with navigator and without navigator seems to be needed. The authors show the efficiency of the navigator with figure 4. But there is no performance (Fidelity, sparsity) comparison between them. If there is the performance comparison, the necessity of navigator can be more convinced.
>
> A: Thanks for your advice. We have added performance comparisons between *with navigator* and *without navigator*. We illustrate the comparisons using both reward-rollout curves and fidelity-sparsity curves in Sec. A.5 and Sec. A.6 in Appendix, respectively. The reward-rollout curve reveals the best solution's fidelity found within a specific rollout number on-the-fly. The fidelity-sparsity curve reveals the fidelity under a specific sparsity threshold after the search. For the reward-rollout comparison, the *with navigator* is above the *without navigator* in most cases, except for the TGAT&Reddit setting (where training the target model and navigator is challenging because of the noisy and imbalanced interactions). The fidelity-sparsity curves are generally consistent with the reward-rollout curves. The *with navigator* achieves a higher fidelity score under varying sparsity thresholds in most cases. We list partial fidelity-sparsity results for TGAT on Wikipedia and Simulate v1 in Table 7 and Table 8, respectively. Note that the performance gaps are more significant on synthetic datasets than those on real-world datasets, as shown in Table 8. The reason is that the target models have better prediction performance on synthetic datasets, which facilitates the learning of our navigator in estimating event importance.
>
> Table 7: Fidelity-sparsity comparison between with navigator and without navigator for explaining TGAT on Wikipedia.
> | Sparsity | Fidelity (without navigator) | Fidelity (with navigator) |
> | :--------: | :--------: | :--------: |
> | 0.2     |2.05|   2.07   |
> | 0.4     | 2.10     |   2.16   |
> | 0.6     | 2.16    |   2.20   |
> | 0.8     | 2.17     |   2.21   |
> | 1.0     | 2.18    |   2.21   |
>
> Table 8: Fidelity-sparsity comparison between with navigator and without navigator for explaining TGAT on Simulate v1.
> | Sparsity | Fidelity (without navigator) | Fidelity (with navigator) |
> | :--------: | :--------: | :--------: |
> | 0.2     | 1.41 |   1.52   |
> | 0.4     | 1.49    |  1.57    |
> | 0.6     | 1.51    |  1.58    |
> | 0.8     | 1.51     |  1.58    |
> | 1.0     |  1.51   |  1.58    |

---

> ### Author Response · Authors · 2022-11-18
> **Response to Reviewer 3, Q3**
>
> >Q3: I think that several experiments are more needed. As far as I can tell, many works for GNN explainers (e.g., D. Luo et al., "Parametrized explainer for graph neural network" at NeurIPS 2020) illustrated the performances, and so that there can be several illustration rather than figure 5 and 6 in this paper?
>
> A: Thanks for your suggestion! We can illustrate the performance from different aspects, not limited to the effectiveness evaluated by the fidelity-sparsity curve. The work *Parametrized explainer for graph neural network*  visualized its explanation result and analyzed the inductive performance and effects of hyperparameters because this work claimed it performs well in the inductive setting. Similarly, we visualized our explanation in Fig. 5 and 6. Then we illustrate the on-the-fly runtime-fidelity / reward-rollout comparison between *with navigator* and *without navigator* in Sec. A.4-6 in Appendix. The reason is that we claim the navigator can speed up searching and help to achieve better results. Finally, we test the effect of hyperparameters in Sec. A.7 in Appendix.

---

> ### Author Response · Authors · 2022-11-18
> **Response to Reviewer 3, Q2**
>
> >Q2: Literature survey would not be enough. The authors claim that it is the first explainer for temporal graph models. But Z. Han et al., "Explainable subgraph reasoning for forecasting on temporal knowledge graphs" at ICLR 2021 tried to construct an explaining model for temporal knowledge graphs. In my view, although it is applied to temporal knowledge graphs, it needs to be mentioned and discussed in the related work section.
>
> A: Thanks for your suggestion.  The work *Explainable subgraph reasoning for forecasting on temporal knowledge graphs* proposed xERTE for temporal knowledge graph reasoning. xERTE achieves self-interpretability by pruning the subgraph used at each inference step based on learned attention scores. The final preserved graph is regarded as an explanation. As there exists a key entity, i.e., predicate, on knowledge graphs, some architecture designs are specific for the predicates. These methods cannot be directly applied to general temporal graphs. However, the interpretation in xERTE is essentially based on attention scores (although there is an extra pruning based on these scores), while we have compared with the attention-based baselines for both target models. Moreover, we have also compared with another self-interpretable temporal neural model named Transformer Hawkes Process (THP), which is based on different mechanisms, i.e., the temporal point process. We use THPExplainer to denote explaining THP using its learned intensity scores. We list Best Fid/AUFSC comparison in Table 6 and fidelity-sparsity curves are in Fig. (13). Briefly, search-based T-GNNExplainer could also outperform the learned THP model's internal self-explanations w.r.t the fidelity-sparsity metric. Detailed results and discussions can be found in Sec. A.8 in Appendix.
>
> Table 6: Best fidelity and AUFSC achieved by T-GNNExplainer and THPExplainer on synthetic datasets.
> |                | Synthetic v1 |        | Synthetic v2 |        |
> |----------------|:------------:|:------:|:------------:|:------:|
> |                |   Best Fid   |  AUFSC |   Best Fid   |  AUFSC |
> | THPExplainer   |     0.127    | -0.485 |     0.207    | -2.046 |
> | T-GNNExplainer |     0.206    | -0.006 |     0.573    |  0.021 |

---

> ### Author Response · Authors · 2022-11-18
> **Response to Reviewer 3, Q1**
>
> > Q1: Several notations are confusing to me. For example, the authors reuse the notation G^i in section 4.3.2, with (minorly) different meaning. But it can be confusing for readers to follow the paper. The reviewer expects that the authors use clear notations to be understood more easily.
>
> A: Thanks for your advice. We have revised these confusing notations and acronyms to make the paper more clear.

---

### Official Review · Reviewer_J8qS · 2022-10-25

**Confidence:** 4
**Correctness:** 3
**Technical Novelty And Significance:** 2
**Empirical Novelty And Significance:** 2
**Recommendation:** 5

**Clarity, Quality, Novelty And Reproducibility:**

The overall writing is good, but there are some minor issues.

The novelty of the proposed idea is somehow limited. The proposed idea uses the common Monte Carlo Tree Search with additional weight learning.

The paper claims to have codes and datasets available. The tuning of hyperparameters should be provided.


**Strength And Weaknesses:**

Strength:
1. The paper proposes an interesting idea of using Monte Carlo Tree Search to select the optimal edge (event) set that explains a prediction.

2. The paper proposes a navigator (multilayer perceptron) to learn Monte Carlo tree weights to speed up the Monte Carlo Tree Search.

3. The paper simulates synthetic datasets by the multivariate Hawkes process and some pre-defined event relation rules. The experiments on the synthetic datasets demonstrate the effectiveness of the proposed method.

Weaknesses:
1. For related works, the paper only covered post-hoc interpretative methods, where the explanatory model is trained on top of the well-trained classification model. However, there are also many works on intrinsically interpretable models, e.g., BrainGNN[1] and dGLCN[2].

2. In experiments, the proposed method has many hyperparameters, but the paper did not provide parameter sensitivity analysis. It will be nice to add hyperparameter tuning results, such as N_{r}, \lambda.

3. Furthermore, considering that the algorithm proposed by the authors is plug-and-play, the paper only validated the idea on two instance-level post-hoc methods, TGAT (2020) and TGN (2020). It will be nice to add experiments on other target models to further demonstrate the effectiveness of the proposed method. There are also some baselines from non-post-hoc explainers (e.g., dGLCN[2]) and model-level algorithms (e.g., XGNN[3]).

4. The overall writing is good, but there are some minor issues. The paper should use the full spelling for the first mention of the acronym, such as GNN in the abstract, and CR on page 6. The authors should use math equation language instead of programming language in Eq. 6.


[1]BrainGNN: Interpretable Brain Graph Neural Network for fMRI Analysis
[2]Dual-Graph Learning Convolutional Networks for Interpretable Alzheimer's Disease Diagnosis
[3]Xgnn: Towards model-level explanations of graph neural networks


**Summary Of The Paper:**

The paper proposed a post-hoc and instance-level explainer called T-GNNExplainer for temporal graph neural network on continuous-time dynamic graphs. The paper provides experimental evaluations on the effectiveness of the proposed explainer on real-world and synthetic datasets.


**Summary Of The Review:**

The paper proposed a post-hoc and instance-level explainer called T-GNNExplainer for temporal graph neural network on continuous-time dynamic graphs. It constructed a navigator and explorer, where the navigator proposed by the paper can play the role of attention to output weights and help the optimization of the Monte Carlo Tree Search. However, the novelty of this paper is not very strong, and more experiment needs to be included. See the strength and weaknesses section for details

---

> ### Author Response · Authors · 2022-11-18
> **Response to Reviewer 2, Q4**
>
> ### Minor issues
> >Q4: The overall writing is good, but there are some minor issues. The paper should use the full spelling for the first mention of the acronym, such as GNN in the abstract, and CR on page 6. The authors should use math equation language instead of programming language in Eq. 6.
>
> A: Thanks for your suggestion. We have changed acronyms and equations carefully in the revised paper.
>
> ### Reference
> [1] Li, Xiaoxiao, et al. "Braingnn: Interpretable brain graph neural network for fmri analysis." Medical Image Analysis 74 (2021): 102233.
>
> [2] Xiao, Tingsong, et al. "Dual-Graph Learning Convolutional Networks for Interpretable Alzheimer’s Disease Diagnosis." International Conference on Medical Image Computing and Computer-Assisted Intervention. Springer, Cham, 2022.
>
> [3] Yuan, Hao, et al. "Xgnn: Towards model-level explanations of graph neural networks." Proceedings of the 26th ACM SIGKDD International Conference on Knowledge Discovery & Data Mining. 2020.
>
> [4] Simiao Zuo, Haoming Jiang, Zichong Li, Tuo Zhao, and Hongyuan Zha. 2020. Transformer Hawkes process. In Proceedings of the 37th International Conference on Machine Learning (ICML'20). JMLR.org, Article 1084, 11692–11702.

---

> ### Author Response · Authors · 2022-11-18
> **Response to Reviewer 2, Q3**
>
> ### More types of target models and explainers
> > Q3: Furthermore, considering that the algorithm proposed by the authors is plug-and-play, the paper only validated the idea on two instance-level post-hoc methods, TGAT (2020) and TGN (2020). It will be nice to add experiments on other target models to further demonstrate the effectiveness of the proposed method. There are also some baselines from non-post-hoc explainers (e.g., dGLCN[2]) and model-level algorithms (e.g., XGNN[3]).
>
> A: Thanks for your valuable advice. We add more types of target models and explainers to analyze our proposed method T-GNNExplainer comprehensively.
>
> In the original submission, we chose TGAT and TGN as target models. TGAT and TGN are both temporal graph-based models. Another mainstream method is the temporal point process, which assumes the happening of events follows a certain point process. Thus, we implement a typical neural point process, THP[4], as a new target model, as stated in the above reply for Q1. We apply T-GNNExplainer to explain THP and compare explanation qualities with the model's intrinsic explanations. We train and explain the new target model on two synthetic datasets because the neural point process requires each edge as a type and they are inapplicable on temporal graphs consisting of tens of thousands of edges (i.e., types).
>
> As for model-level explainers, there are few works even for static graphs. XGNN is the typical method. More importantly, the explanation output of model-level and instance-level explainers are markedly different. Model-level explainers aim to output a single prototype for a model, while instance-level explainers focus on influential elements for a specific prediction on a local subgraph. They are not directly comparable and there do not exist commonly used evaluation metrics to compare model-level and instance-level explainers as well. We believe the model-level explainer for temporal graphs is still an unsolved problem needed much time to explore. Our instance-level explainer may provide some insights for model-level explainers on temporal graphs in the future.

---

> ### Author Response · Authors · 2022-11-18
> **Response to Reviewer 2, Q2**
>
> ### Hyperparameters
> >Q2: In experiments, the proposed method has many hyperparameters, but the paper did not provide parameter sensitivity analysis. It will be nice to add hyperparameter tuning results, such as $N_{r}$, $\lambda$.
>
> A: Thanks for your advice. We have added a hyperparameter analysis on $\lambda$. The hyperparameter $\lambda$ in Eq.(4) balances the exploration and exploitation in searching. Hence it influences the quality of searched results. As for $N_{r}$, it can be determined by the target models, as these target models mainly stack two layers, and each layer aggregate 6~8 temporal edges. After some exploratory experiments and statistics, we set the $N_{r}$ to 25 by default since it could cover most of the temporal events used by target models. Hence, we mainly analyze the effect of $\lambda$. We evaluate two target models on the Wikipedia dataset with four $\lambda$ settings, i.e, 1, 5, 10, and 100. Results in Fig. (12) reveal that a smaller $\lambda$ generally outperforms a larger one. Because the navigator has estimated the candidate's importance, exploiting is preferred over exploring in our framework. We also list results for TGN&Wikiprdia in Table 5. Although a smaller $\lambda$ is slightly better, the absolute difference is not very significant. In practice, a number in the range [1, 10] is recommended. Detailed results and discussions can be found in Sec. A.7 in Appendix.
>
> Table 5: Effect of $\lambda$ for explaining TGN on Wikipedia. Fidelity (i) indicates $\lambda$ is set to i.
> | Sparsity | Fidelity (1) | Fidelity (5) | Fidelity (10) | Fidelity (100) |
> | :--------: | :--------: | :--------: | :--------: | :--------: |
> | 0.2     |   0.461   |  0.455    |  0.456    |  0.455    |
> | 0.4     |   0.622   |  0.611    |  0.613    |  0.611    |
> | 0.6     |   0.713   |  0.705    |  0.706    |  0.705    |
> | 0.8     |   0.755   |  0.747    |  0.748    |  0.747    |
> | 1.0     |   0.779   |  0.770    |  0.771    |  0.770    |

---

> ### Author Response · Authors · 2022-11-18
> **Response to Reviewer 2, Q1**
>
> ### Intrinsically Interpretable models
> > Q1: For related works, the paper only covered post-hoc interpretative methods, where the explanatory model is trained on top of the well-trained classification model. However, there are also many works on intrinsically interpretable models, e.g., BrainGNN[1] and dGLCN[2].
>
> A: Thanks for the related works provided. The works BrainGNN and dGLCN utilize some ***intrinsically interpretable designs*** for ***biomedical tasks***. BrainGNN aims to conduct static graph-level fMRI-related classification tasks. The interpretation is achieved by discovering important nodes (i.e., salient regions of the brain) based on a specific pooling layer. dGLCN focuses on Alzheimer’s disease (AD) diagnosis using subject graphs and feature graphs. The interpretation is reflected by the learned subject and feature correlation importance. However, the model design and interpretation of these works are tightly coupled with the data characteristics in their tasks. For BrainGNN, brain region-aligned graphs are required. For dGLCN, the node features correspond to specific region-of-interests as well. Moreover, all these works do not consider temporal information, which is the main concern in our work. Hence, we consider that these works cannot be compared with our method straightforwardly.
>
> However, it is a good idea to consider the intrinsically interpretable values in models as baselines to show the (dis)advantages of our proposed method T-GNNExplainer. Here we have two methods for temporal graphs, i.e., ***Attention*** and ***THPExplainer***. BrainGNN and dGLCN adopted the learned model weights as explanations, to represent which brain region is activated, or which subject/feature is highly correlated to the prediction. Similarly, we used the learned attention score as an explanation, to represent the importance of previous events to the target event. Please note that this **Attention** baseline is already implemented in our original submission.
>
> Moreover, we added the **THPExplainer** inspired by some previous works using temporal point processes to model temporal events. These models are somewhat self-interpretable based on their learned intrinsic conditional intensity scores, which describe the positive or negative influence of past events on future ones. Thus, the learned intensity can be regarded as an explanation to represent the event's importance. In detail, we added a target model, i.e., **Transformer Hawkes Process (THP)**, and devised the corresponding explainer, i.e., **THPExplainer**. THPExplainer utilizes intrinsic intensity scores of THP to conduct explanations. We compare T-GNNExplainer with THPExplainer to reveal the explanation quality difference. Full results are shown in Fig. (13). For conciseness, we only list partial results on Simulate v1 in Table 3 and Best Fid/AUFSC in Table 4. The results show that the TGNNExplainer could even achieve higher fidelity scores than merely using the learned intensity in these neural point process models. Detailed results and analysis can be found in Sec. A.8 in Appendix.
>
>
> Table 3: Fidelity-Sparsity comparison between T-GNNExplainer and THPExplainer on Simulate v1.
> | Sparsity | Fidelity (THPExplainer) | Fidelity (T-GNNExplainer) |
> | :--------: | :--------: | :--------: |
> | 0.2     | -0.977     | -0.267     |
> | 0.4     | -0.768     | -0.002     |
> | 0.6     | -0.330     | 0.097     |
> | 0.8     | 0.055     | 0.177     |
> | 1.0     | 0.127     | 0.206     |
>
>
> Table 4: Best fidelity and AUFSC achieved by T-GNNExplainer and THPExplainer on synthetic datasets.
> |                | Synthetic v1 |        | Synthetic v2 |        |
> |----------------|:------------:|:------:|:------------:|:------:|
> |                |   Best Fid   |  AUFSC |   Best Fid   |  AUFSC |
> | THPExplainer   |     0.127    | -0.485 |     0.207    | -2.046 |
> | T-GNNExplainer |     0.206    | -0.006 |     0.573    |  0.021 |

---

### Official Review · Reviewer_3TUm · 2022-10-25

**Confidence:** 4
**Correctness:** 3
**Technical Novelty And Significance:** 3
**Empirical Novelty And Significance:** 3
**Recommendation:** 6

**Clarity, Quality, Novelty And Reproducibility:**

Overall, the paper's presentation is clear and easy to understand. There is a bit issue in terms of the novelty of the proposed model.

**Strength And Weaknesses:**

Strengths:
- The proposed method is one of the first graph explainer models for temporal graphs.
- The combination of the MTCS explorer and the navigator is interesting.
- The author shows the performance benefit on temporal graph explanation in terms of fidelity and sparsity.

Weakness:
- The idea of using parameterized navigator as well as MTCS explorer has been proposed in the previous graph explainers.
- The use of MTCS in the solution search makes the running time relatively slow. The authors have not discussed the runtime complexity as well as the runtime comparison with the baselines. In static graphs PGExplainer runs much faster (in milliseconds). It will be good to see the runtime-performance trade-off as PGExplainer can also be modified to handle temporal graphs as well.
- There is related work on the temporal graph explainer that was submitted to arXiv not long before the ICLR deadline, which may indicate concurrent works. However, it would still be beneficial if the author could compare and contrast with the work.
> He, Wenchong, Minh N. Vu, Zhe Jiang, and My T. Thai. "An Explainer for Temporal Graph Neural Networks." arXiv preprint arXiv:2209.00807 (2022).

**Summary Of The Paper:**

The author proposed a new method for explaining temporal graph models. Most of the previous explainer models only focus on static graphs. To achieve the goal, the authors combine the explorer framework that uses Monte Carlo Tree Search with navigator frameworks that guide the search by predicting the correlation between an event to the target event. The authors then demonstrate the benefit of the proposed model in real-world and synthetic data.

**Summary Of The Review:**

The paper has a nice contribution to temporal graph explainability. However, there are a few weaknesses, particularly in terms of the novelty of the proposed method.
Based on the considerations above, I recommend a weak acceptance.

---

> ### Author Response · Authors · 2022-11-18
> **Response to Reviewer 1, Q3**
>
> > Q3: There is related work on the temporal graph explainer that was submitted to arXiv not long before the ICLR deadline, which may indicate concurrent works. However, it would still be beneficial if the author could compare and contrast with the work. He, Wenchong, Minh N. Vu, Zhe Jiang, and My T. Thai. "An Explainer for Temporal Graph Neural Networks." arXiv preprint arXiv:2209.00807 (2022).
>
> A: Thanks for providing one related work. The work *An Explainer for Temporal Graph Neural Networks* is based on a sequence of static snapshots of a temporal graph. It leverages PGM-explainer to explain each snapshot and a special algorithm to discover dominant bayesian networks. We consider that our work differs from this work in both problem formulation and optimization objectives. We do not split the temporal graph into snapshots as the sequence of static snapshots is essentially a very coarse approximation of the actual temporal graph where interactions happen continuously. How to optimally split the graph is non-trivial in different datasets, which may lead to a significant loss of information. Instead, we regard the graphs as sequences of temporal events so that the information of the finest granularity is kept. Besides, the mentioned work mainly focuses on discovering bayesian relations by using a temporal bayesian information criterion, while we aim to maximize the mutual information between the explanation and the original model prediction. Hence, we concern that it may be problematic to compare ours with this work directly. However, we added and discussed this work in the related work section.

---

> ### Author Response · Authors · 2022-11-18
> **Response to Reviewer 1, Q2**
>
> >Q2: The use of MTCS in the solution search makes the running time relatively slow. The authors have not discussed the runtime complexity as well as the runtime comparison with the baselines. In static graphs PGExplainer runs much faster (in milliseconds). It will be good to see the runtime-performance trade-off as PGExplainer can also be modified to handle temporal graphs as well.
>
> A: We have a runtime comparison in Sec. A.4 in the original submission. Moreover, we extended efficiency experiments and added a complexity analysis. Table 1 below are results listed in Table 5 in the paper. Results show that although T-GNNExplainer is slower than non-search based baselines, considering that the AUFSC is much higher than other baselines, the running time of T-GNNExplainer with navigator is acceptable in most cases. For the complexity analysis, since MCTS is an anytime algorithm, i.e., it can stop at any time based on the computational budget, the complexity is O(NDC). N is the number of rollouts, D is the expansion depth of each rollout determined by the sparsity threshold, and C is a constant including inference time of the navigator, storing time of tree nodes, and other constant-time operations. **Moreover, we added runtime-fidelity curves in Sec. A.4 to illustrate the tradeoff between efficiency and solution quality more intuitively** **(For conciseness, we only list results for TGAT on wikipedia in the following Table 2)**. Although the runtime increases with the rollout number increases, the fidelity grows steeply at the very beginning stage and slowly in subsequent  iterations. Hence in practice, we may set a relatively small rollout number (e.g., 100) to achieve an appropriate tradeoff.
>
> Table 1: Running time comparison of different methods on all the datasets for explaining an instance with TGAT. † indicates withholding the navigator.
> |                  | Methods/Time (s) | Wikipedia | Reddit | Synthetic v1 | Synthetic v2 |
> |:----------------:|:----------------:|:---------:|:------:|:------------:|:------------:|
> | non-search based |       ATTN       |    0.05   |  0.17  |     0.05     |     0.05     |
> |                  |       PBONE      |    0.31   |  0.39  |     0.23     |     0.25     |
> |                  |        PG        |    0.03   |  0.22  |     0.03     |     0.03     |
> |   search based   |  T-GNNExplainer† |   103.4   |  158.2 |     89.74    |     178.2    |
> |                  |  T-GNNExplainer  |   20.38   |  28.2  |     14.49    |     12.5     |
>
> Table 2: Runtime-fidelity tradeoff for the TGAT model on Wikipedia.
> | Rollout | Runtime (s) | #MCTS Nodes | Fidelity | (Current fidelity)/ (Max fidelity) |
> | :--------: | :--------: | :--------: | :--------: | :--------: |
> | 50     |  11.7    |   584   | 1.71 | 0.77 |
> | 100     |  22.8    |  1139    | 1.89 | 0.85 |
> | 150     |   33.6   |  1677    | 1.96 | 0.88 |
> | 200     |   44.2   |  2206    | 2.05 | 0.92 |
> | 300     |   64.9   |  3239    | 2.14 | 0.97 |
> | 400     |   84.6   |  4226    | 2.17 | 0.98 |
> | 500     |   103.4   |  5172    | 2.10 (Max fidelity) | 1.00 |

---

> ### Author Response · Authors · 2022-11-18
> **Response to Reviewer 1, Q1**
>
> > Q1: The idea of using parameterized navigator as well as MTCS explorer has been proposed in the previous graph explainers.
>
> A: Thanks for your question. Although both methods have been proposed in previous works, our work differs from them in several ways. Firstly, these methods are generally adopted to explain static graph data. In this work, we focus on temporal graphs, where special considerations should be taken on the time. Secondly, a parameterized module in previous works is used as a complete explanation method. Instead, in our work, we only utilize it as a learnable module to output significance estimations of candidates on-the-fly and facilitate the search process. The module injects both spatial and temporal information into searching events. To the best of our knowledge, introducing pre-trained modules into MCTS for explanations is novel.

---

### Author Response · Authors · 2022-11-18
**To all reviewers**

Thanks for all reviewers' constructive advice and comments, we summarize major changes as follows:
1. We revised the paper for some expression issues, including confusing notations, acronyms, and equations, etc.
2. We revised the related work section to include and discuss more related works.
3. We extended the runtime comparison in Sec. A.4 to compare the proposed method's efficiency with baselines. We added a complexity analysis as well.
4. We added two sections, i.e., Sec. A.5 and A.6, to investigate the performance effect of the navigator module more thoroughly.
5. We added Sec. A.6 to discuss the effect of an important hyperparameter $\lambda$ in Eq.4 in the paper.
7. As suggested by reviewers, we considered one more type of target model and explainer. Except for TGAT and TGN as temporal graph-based models, we added the Transformer Hawkes Process (THP) as a new target model, which is based on the temporal point process. Further, we compare with a  specifically devised THPExplainer utilizing THP's internal self-interpretable scores. Results are described in Sec. A.8.

Please note that:
1. All revisions are highlighted in red.
2. The figures mentioned below refer to those in the paper and the tables refer to those embedded in the reply.


In the following, we respond to each reviewer's concerns one by one.

---

### Author Response · Authors · 2022-12-01
**A kindly reminder for feedbacks**

Dear reviewers,

We have made effects to polish the paper and try our best to meet the concerns of all comments in the previous round's review. We sincerely anticipate constructive and active discussions with reviewers/AC.

Thanks for your consideration!

---

### Decision · Program_Chairs · 2023-01-20

**Decision:**

Accept: poster

**Justification For Why Not Higher Score:**

There are still some remaining issues, e.g. regarding writing quality and possibly additional experiments such as real world case studies.

**Justification For Why Not Lower Score:**

The paper is well-motivated, and provides good performance on the novel and interesting problem of explaining temporal graphs.

**Metareview: Summary, Strengths And Weaknesses:**

The paper proposes the "T-GNNExplainer" approach which aims to explain the predictions of temporal graph models, specifically; given a target event, we want to find a subset of temporal events that are most relevant to a particular model prediction. T-GNNExplainer involves an "explorer" to efficiently find event subsets using Monte Carlo Tree Search; and a navigator framework to guide the search by predicting correlations to the target event.

In general, reviewers appreciated the method's novelty as one of the first graph explainers for temporal graphs, the interesting design of the MCTS explorer and navigator, and the strong performance in experiments. For the experiments, some issues were raised by reviewers such as lack of hyperparameter sensitivity analysis, and discussion of related work (e.g. xERTRE) and adding additional baselines, and revising the notation to improve the clarity.

The raised issues were successfully addressed by the authors, who added a range of different improvements, particularly, to the clarity of writing, adding related work and additional baselines to the experiments, running time, ablations, and some further experiments such as related to the fidelity-sparsity comparison. The additions were appreciated by the reviewers, who generally felt that all their concerns had been well-addressed. I thank the authors for their efforts in improving the quality of the paper.

In the end, reviewers and AC find that the paper is well-motivated, and provides good performance on the novel and interesting problem of explaining temporal graphs. Hence, I recommend acceptance.

**Note From Pc:**

if the above contains the word "oral" or "spotlight" please see: "oral" presentation means -> notable-top-5% and "spotlight" means -> notable-top-25%. As stated in our emails, we are disassociating presentation type from AC recommendations